# Retro-fallback: retrosynthetic planning in an uncertain world

**Austin Tripp**[1]* **Krzysztof Maziarz**[2] **Sarah Lewis**[2]
**Marwin Segler**[2] **José Miguel Hernández-Lobato**[1]
[1]University of Cambridge   [2]Microsoft Research AI4Science
{ajt212,jmh233}@cam.ac.uk
{krmaziar,sarahlewis,marwinsegler}@microsoft.com

## Abstract

Retrosynthesis is the task of proposing a series of chemical reactions to create a
desired molecule from simpler, buyable molecules. While previous works have
proposed algorithms to find optimal solutions for a range of metrics (e.g. shortest,
lowest-cost), these works generally overlook the fact that we have imperfect knowl-
edge of the space of possible reactions, meaning plans created by the algorithm may
not work in a laboratory. In this paper we propose a novel formulation of retrosyn-
thesis in terms of stochastic processes to account for this uncertainty. We then pro-
pose a novel greedy algorithm called retro-fallback which maximizes the probability
that at least one synthesis plan can be executed in the lab. Using in-silico bench-
marks we demonstrate that retro-fallback generally produces better sets of synthesis
plans than the popular MCTS and retro* algorithms. We encourage the reader to
view the full version of this paper at `https://arxiv.org/abs/2310.09270`.

## 1  Introduction

Retrosynthesis (planning the synthesis of organic molecules via a series of chemical reactions) is
a common task in chemistry with a long history of automation (Vleduts, 1963; Corey and Wipke,
1969). Although the combinatorially large search space of chemical reactions makes naive brute-force
methods ineffective, recently significant progress has been made by developing modern machine-
learning based search algorithms for retrosynthesis (Strieth-Kalthoff et al., 2020; Tu et al., 2023).
However, there remain obstacles to translating the output of retrosynthesis algorithms into real-world
syntheses. One significant issue is that these algorithms have imperfect knowledge of the space
of chemical reactions. Because the underlying physics of chemical reactions cannot be efficiently
simulated, retrosynthesis algorithms typically rely on data-driven reaction prediction models which
can "hallucinate" unrealistic outputs (Zhong et al., 2023), akin to hallucinated outputs in other
domains (OpenAI, 2023). This results in synthesis plans which cannot actually be executed.

Although future advances in modelling may reduce the prevalence of infeasible reactions, we think
it is unlikely that they will ever be eliminated entirely, as even the plans of expert chemists do not
always work on the first try. One possible workaround to failing plans is to produce *multiple* synthesis
plans instead of just a single one: the other plans can act as *backup* plans in case the primary plan
fails. Although existing algorithms may find multiple synthesis plans, they are generally not designed
to do so, and there is no reason to expect the plans found will be suitable as *backup* plans (e.g. they
may share steps with the primary plan and thereby fail alongside it).

In this paper, we present several advancements towards retrosynthesis with backup plans. First, in
section 3 we explain how uncertainty about whether a synthesis plan will work in the wet lab can

---

*Work done partly during internship at Microsoft Research AI4Science

NeurIPS 2023 AI for Science Workshop.

be quantified with stochastic processes. We then propose an evaluation metric called *successful synthesis probability* (SSP) which quantifies the probability that *at least one* synthesis plan found by an algorithm will work. This naturally captures the idea of producing backup plans. Second, in section 4 we present a novel search algorithm called *retro-fallback* which greedily optimizes SSP, and explain qualitatively how it avoids potential failure modes of other algorithms. Finally, in section 6 we demonstrate quantitatively that retro-fallback outperforms existing algorithms on an in-silico benchmark. Together, we believe these contributions form a notable advancement towards translating results from retrosynthesis algorithms into the lab. Note that this is an abbreviated version of a paper under review of another venue. The full version will be linked to at a later time.

## 2 Background: what is retrosynthesis?

Let $\mathcal{M}$ represent the space of molecules and $\mathcal{R}$ represent the space of reactions, where each reaction transforms a set of *reactant* molecules in $2^{\mathcal{M}}$ into a *product* molecule in $\mathcal{M}$. Retrosynthesis is usually formalized as a search problem on a graph $\mathcal{G}$, defined implicitly via a *backward reaction model* $B : \mathcal{M} \mapsto 2^{\mathcal{R}}$ which defines all possible reactions for a given molecule. Because $\mathcal{G}$ is combinatorially large, most search algorithms only store a small explicit subgraph $\mathcal{G}' \subseteq \mathcal{G}$. We refer to nodes which may have children in $\mathcal{G}$ but have no children in $\mathcal{G}'$ as *tip* nodes.[2] In general, search algorithms alternate between selecting tip nodes in $\mathcal{G}'$ and querying $B$ to add new nodes to $\mathcal{G}'$ until the computational budget is exhausted (a process called *expansion*).

There are multiple ways to define the nodes and edges of $\mathcal{G}$. We choose $\mathcal{G}$ to be an *AND/OR graph*: a directed graph containing nodes for molecules and reactions. Edges exist only from reactions to their reactant molecules and from molecules to reactions that produce them, making $\mathcal{G}$ bipartite.[3] Examples of AND/OR graphs are shown in Figure 1. Reactions can be naturally associated with "AND nodes" because *all* of their reactant molecules must be synthesized for the reaction to work, while molecules are associated with "OR nodes" because *any* reaction can be used to synthesize it. However, to avoid confusion we will simply refer to the nodes of $\mathcal{G}$ as molecules and reactions.

Given a *target molecule* $m_t \in \mathcal{M}$, the primary goal of retrosynthesis algorithms is to find *synthesis plans*: subtrees $T \subseteq \mathcal{G}$ rooted at $m_t$ containing at most one reaction to produce each molecule. For these plans to be executable, all tip nodes of $T$ must be contained in an *inventory* $\mathcal{I} \subseteq \mathcal{M}$ of buyable molecules. Among all synthesis plans, those with minimum cost or highest quality are preferred, commonly formalized with a scalar cost/value function (Segler et al., 2018; Chen et al., 2020).

## 3 A formulation for retrosynthesis with uncertainty

### 3.1 Stochastic processes over reaction uncertainty

To account for synthesis plans not working in the lab, we must first define what it means for a synthesis plan to "work". As mentioned in the introduction, the most obvious failure mode is that one of the reactions in the plan cannot be performed. This could happen for a variety of reasons: it may not produce the desired product, produce dangerous by-products, have a low yield, or require specialized expertise or equipment. Rather than trying to explicitly model these factors, we propose to collapse all nuance into a binary outcome: a reaction is either *feasible* or *infeasible*. Not only is this easier to model, we note that ultimately if a chemist performs a reaction they will either move to the next step in the synthesis plan or admit defeat and abandon the synthesis plan (a binary outcome). Therefore, for a given chemist and lab, we postulate the existence of a binary "feasibility" function $f^* : \mathcal{R} \mapsto \{0, 1\}$ which we will use to create and evaluate synthesis plans.

A second reason why a synthesis plan may not work is the inability to buy one of the starting molecules. This is usually not a significant issue since vendors update their inventories in real time and offer fast delivery, especially for common chemicals. However, this is not always the case: for example, some companies offer large "virtual libraries" with billions of molecules which they *believe* they can synthesize upon request, but not with 100% reliability. To account for this, we therefore define a binary "buyability" function $b^* : \mathcal{M} \mapsto \{0, 1\}$ analogously to $f^*$.

---

[2]In contrast with *leaf* nodes which have no children in $\mathcal{G}$, e.g. molecules where no reactions are possible.

[3]Note that in *retro*synthesis, the edges go in the *opposite* direction of the chemical reactions.

If we knew $f^*$ and $b^*$ then retrosynthesis would simply be a search problem (albeit a large one). However, in practice they are unknown. A natural response is therefore to model our *epistemic uncertainty* about $f^*$ and $b^*$. One approach is to model point-wise uncertainties, using some mechanism to predict $P(f^*(r) = 1)$ and $P(b^*(m) = 1)$ for all $r$ and $m$. Unfortunately, this approach is fundamentally incapable of capturing beliefs about *correlations* between different outcomes. Instead, we propose to model uncertainty about $f^*$ and $b^*$ directly in function space using *stochastic processes* (essentially distributions over functions). We define a *feasibility model* $\xi_f$ to be a binary stochastic process over $\mathcal{R}$, and define a *buyability model* $\xi_b$ to be a binary stochastic process over $\mathcal{M}$. This model class is very general: the fully-deterministic formulation from is a special case where $\xi_f$ and $\xi_b$ are degenerate distributions[4], while independent outcomes are a special case where $f/b$ are independent Bernoulli random variables at all points. Other more interesting stochastic processes which induce correlations could be constructed by putting a prior over the parameters of a model (e.g. Bayesian neural networks (MacKay, 1992)) or using non-parametric models like Gaussian processes (Williams and Rasmussen, 2006).

## 3.2 New evaluation metric: successful synthesis probability

Given $f$ and $b$, a successful synthesis plan $T \subseteq \mathcal{G}$ must have $f(r) = 1$ for all $r \in T$ and $b(m) = 1$ for every tip molecule in $T$. However, if we are uncertain about $f$ and $b$ then the distinction between "successful" and "unsuccessful" synthesis plans is not binary: every $T$ could have some probability of succeeding. What then should be the goal of retrosynthesis?

There is no objectively correct answer to this question. Although one could try to find the synthesis plan with the highest probability of succeeding, we instead propose the goal of maximizing the probability that *any* synthesis plan $T \subseteq \mathcal{G}'$ is valid. Assuming that a chemist would be willing to try all the synthesis plans in $\mathcal{G}'$, this goal not only captures the spirit of finding good synthesis plans, but backup plans too. Specifically, let $s(m; \mathcal{G}', f, b)$ represent the *successful synthesis* of a molecule $m$: 1 if $m$ can be bought or synthesized using only feasible reactions in $\mathcal{G}'$, otherwise 0. We write $s(m)$ when $\mathcal{G}', f, b$ are clear from context. If $s(m_t; \mathcal{G}', f, b) = 1$, this implies there is a successful synthesis plan for the target molecule. We then define the *successful synthesis probability* (SSP) as

$$\bar{s}(m; \mathcal{G}', \xi_f, \xi_b) = P_{f \sim \xi_f, b \sim \xi_b}[s(m; \mathcal{G}', f, b) = 1] \tag{1}$$

and propose using SSP to evaluate the success of retrosynthesis algorithms. Figure 1 illustrates this and contrasts it with the traditional binary view of retrosynthesis, wherein every node is either "solved" or "unsolved". The formulation with stochastic processes implies a non-binary degree of "solvedness" for each node, representing the fraction of scenarios where each node is solved. SSP is the "solvedness" of the root node.

## 3.3 Computing successful synthesis probability

Unfortunately, SSP is not easy to compute exactly: we prove in Appendix B.1 that it is NP-hard to compute (the proof essentially shows that a known NP-hard problem can be formulated as calculating SSP in a suitably-chosen graph). Although this result may appear to show that SSP is not a practical evaluation metric, it does not preclude the existence of an efficient randomized algorithm to *estimate* SSP. This is exactly what we propose. First, note that given $f/b$, if we define a similar concept of success for reactions then s can be defined recursively in terms of its children in $\mathcal{G}'$ (provided by the function $Ch_{\mathcal{G}'}$):

$$s(m; \mathcal{G}', f, b) = \max\left[b(m), \max_{r \in Ch_{\mathcal{G}'}(m)} s(r; \mathcal{G}', f, b)\right] \tag{2}$$

$$s(r; \mathcal{G}', f, b) = f(r) \prod_{m \in Ch_{\mathcal{G}'}(r)} s(m; \mathcal{G}', f, b). \tag{3}$$

This suggests that dynamic programming can be used to compute $s(m_t; \mathcal{G}', f, b)$ in polynomial time (we prove this in Appendix B.2). Second, observe that if $f \sim \xi_f, b \sim \xi_b$ then $s(m_t; \mathcal{G}', f, b)$ is a Bernoulli random variable with mean $\bar{s}(m_t; \mathcal{G}', \xi_f, \xi_b)$. This suggests a natural estimator:

$$\hat{s}(m_t; \mathcal{G}', \xi_f, \xi_b, k) = \frac{1}{k} \sum_{i=1}^{k} s(m_t; \mathcal{G}', f_k, b_k) \qquad f_1, \ldots, f_k \sim \xi_f, \quad b_1, \ldots, b_k \sim \xi_b. \tag{4}$$

---

[4]Specifically, $\xi_f$ contains only $f(r) = \mathbf{1}_{\exists m: r \in B(m)}$ and $\xi_b$ contains only $b(m) = \mathbf{1}_{m \in \mathcal{I}}$.

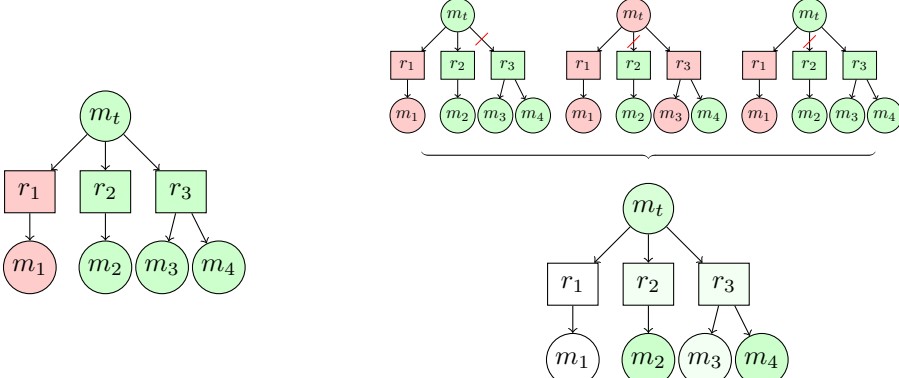

Figure 1: Illustration of retrosynthesis with and without uncertainty. **Left:** Traditional binary formulation: molecules are either buyable or not. All synthesis routes which use only buyable molecules are "solutions." Here, $m_t$ is solved via both $r_2$ and $r_3$. **Right:** Formulation with stochastic processes, where each reaction may fail and molecules may or may not be buyable. Each sample from $\xi_f, \xi_b$ implies that each node is either solved or unsolved (top). Averaging over these samples produces a non-binary "solvedness" value for each node, indicated in shades of green (bottom).

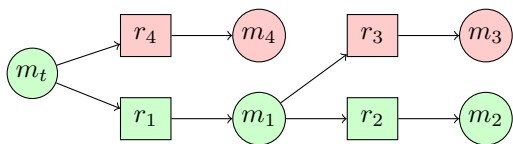

Figure 2: AND/OR graph illustrating how maximizing SSP can be different from finding individually successful synthesis plans (cf. 4.1). Green nodes are part of a synthesis plan, red nodes are not.

## 4 Retro-fallback: a greedy algorithm to maximize SSP

### 4.1 Existing algorithms can fail to maximize SSP

Before introducing a new algorithm, we examine what shortcomings (if any) a new algorithm should be designed to overcome. In theory, there is nothing preventing existing algorithms from effectively maximizing SSP. One could run any algorithm *independently* of $\xi_f, \xi_b$, compute SSP *post hoc*, and potentially get a high value. However, to try to *ensure* a high SSP value, it is natural to explicitly try to maximize SSP by suitably configuring the "objective" of existing algorithms. Unfortunately, existing algorithms do not have an "objective" which can be set arbitrarily. For example, retro* has an independent cost for each reaction and molecule (Chen et al., 2020), MCTS uses a reward function for individual plans (Segler et al., 2018), while algorithms like breadth-first search or proof-number search (Kishimoto et al., 2019) have no customizable rewards or costs of any kind. Because SSP depends on an entire graph and distributions over reaction feasibilities and molecule buyabilities (which may involve correlations), we think *it is not generally possible to set SSP as the objective of previously-proposed retrosynthesis algorithms*. For most algorithms, we believe the closest proxy for maximizing SSP is to optimize for individually successful synthesis plans, or plans containing individually feasible reactions and buyable molecules.

Although these objectives may seem similar, it is not difficult to imagine cases where they differ. Figure 2 illustrates such a case, wherein a synthesis plan with reactions $r_1, r_2$ has been found and the algorithm must choose between expanding $m_3$ or $m_4$. Individually either molecule could be promising, but any new synthesis route proceeding via $r_3$ will also use $r_1$ and is therefore prone to fail alongside the existing plan if $r_1$ turns out to be infeasible. Even though $m_4$ may not be the best choice in general, an algorithm maximizing SSP would clearly need to account for the interaction between existing and prospective synthesis plans in its decision making, which simply is not possible by reasoning about individual synthesis plans in isolation. This provides compelling motivation to develop algorithms which account for interactions between synthesis plans.

## 4.2 Ingredients for an informed, greedy search algorithm

A natural starting point for an algorithm specifically designed to maximize SSP is to estimate how different actions might affect SSP, and choose actions accordingly. Theorem B.1 suggests that computing this exactly will scale poorly to larger search graphs, and therefore we assert that the basis of any efficient algorithm must be *samples* from $\xi_f, \xi_b$. Furthermore, equations 2–3 show how the success of any given node in $\mathcal{G}'$ can be computed reasonably efficiently in terms of the success values of other nodes. This broadly suggests it might be possible to modify equations 2–3 to perform *counterfactual reasoning*: i.e. predicting what $s(m_t; \mathcal{G}', f, b)$ *could be* if $\mathcal{G}'$ were modified.

We take inspiration from the greedy retro* algorithm (Chen et al., 2020) and instead consider the counterfactual of simultaneously expanding all tip nodes on an *entire synthesis plan* $T \subseteq \mathcal{G}'$. For a tip molecule $m$, we have that $s(m; \mathcal{G}, f, b) = b(m)$ from equation 2. If $b(m) = 1$ then expanding $m$ cannot change $s(m)$ (it is already at its maximum value), but if $b(m) = 0$ then it is possible that $s(m)$ will change to 1 upon expansion. This is a natural entry point for a *search heuristic*: let $h : \mathcal{M} \mapsto [0, 1]$ be a heuristic function mapping molecules to probabilities that $s(m)$ will become 1 upon expansion (assuming $b(m) = 0$).

We then assume that expanding a synthesis plan $T$ amounts to setting $s(m) = 1$ for all non-buyable tip nodes $m \in T$ with $s(m) = 0$ *independently* with probability $h(m)$. For any $m \in \mathcal{G}'$, we define $\psi(m; \mathcal{G}', f, b, h)$ to be the largest expected value of $s(m)$ obtained across synthesis plans $T \subseteq \mathcal{G}'$ for $m$ under the expansion scenario above and use $\psi(m)$ when $\mathcal{G}', f, b, h$ are clear from context. Critically, because all tip nodes are considered independently under the expansion process, the optimal plan for every molecule $m$, that is, the plan that maximizes the expectation of $s(m)$ under the expansion of its tip nodes, will consist of sub-plans which are also individually optimal. This allows us to define $\psi$ implicitly with the recurrence relation:

$$\psi(m; \mathcal{G}', f, b, h) = \begin{cases} \max\left[b(m), h(m)\right] & \text{(Tip molecule)} \\ \max\left[b(m), \max_{r \in Ch_{\mathcal{G}'}(m)} \psi(r; \mathcal{G}', f, b, h)\right] & \text{(Non-tip molecule)} \end{cases} \quad (5)$$

$$\psi(r; \mathcal{G}', f, b, h) = f(r) \prod_{m \in Ch_{\mathcal{G}'}(r)} \psi(m; \mathcal{G}', f, b, h) . \quad (6)$$

Equations 5–6 effectively propagate information "up" the graph towards $m_t$, until $\psi(m_t)$ contains a useful quantity: the expected value of $s(m_t)$ upon optimal expansion. A similar technique can be used to propagate this information "down" the graph towards the tip nodes to decide which tip node(s) to expand. For any $m \in \mathcal{G}'$, let $\rho(m; \mathcal{G}', f, b, h)$ represent the maximum expected value of $s(m_t)$ upon expansion across all synthesis plans $T \subseteq \mathcal{G}'$ that contain both $m_t$ and $m$. We call a plan achieving $\rho(m; \mathcal{G}', f, b, h)$ a *constrained* optimal plan for $m$, while a plan achieving $\psi(m; \mathcal{G}', f, b, h)$ is called an *unconstrained* optimal plan for $m$. The assumption of independence implies that the constrained optimal plan for node $n$ can be formed by replacing part of the constrained optimal plan for the parent of $n$ with the unconstrained optimal plan for $n$, leading to the following recursive definition of $\rho$:

$$\rho(m; \mathcal{G}', f, b, h) = \begin{cases} \psi(m; \mathcal{G}', f, b, h) & m \text{ is root node} \\ \max_{r \in Pa_{\mathcal{G}'}(m)} \rho(r; \mathcal{G}', f, b, h) & \text{all other } m \end{cases} \quad (7)$$

$$\rho(r; \mathcal{G}', f, b, h) = \begin{cases} 0 & \psi(r; \mathcal{G}', f, b, h) = 0 \\ \rho(Pa_{\mathcal{G}'}(r)) \frac{\psi(r)}{\psi(Pa_{\mathcal{G}'}(r))} & \psi(r; \mathcal{G}', f, b, h) > 0 \end{cases} \quad (8)$$

Here, $Pa_{\mathcal{G}'}$ yields node's parents in $\mathcal{G}'$. The last remaining question is how $\psi$ and $\rho$ can be computed. If $\mathcal{G}'$ is acyclic then $\psi$ and $\rho$ can be calculated in linear time by iterating equations 5–6 from tip nodes to the root, then iterating equations 7–8 from root to tip nodes. If there are cycles then the cost could potentially be larger, but in Appendix B.2 we prove it is at most quadratic. In any case, it is clear that $\psi$ and $\rho$ can form the basis for an efficient search algorithm.

## 4.3 Retro-fallback: a full greedy algorithm

Creating a full greedy algorithm requires aggregating information across many samples from $\xi_f$ and $\xi_b$ to decide which tip node to expand at each step. Recalling our motivation of producing synthesis plans with backup plans, we propose to greedily expand molecules which are predicted to form

---
**Algorithm 1** Retro-fallback algorithm
---
**Require:** target molecule $m_t$, max iterations $L$, backward reaction model $B$, search heuristic $h$
**Require:** samples $f_1, \ldots, f_k \sim \xi_f$, $b_1, \ldots, b_k \sim \xi_b$
  1: $\mathcal{G}' \leftarrow m_t$
  2: **for** $i$ in $1, \ldots, L$ **do**
  3:     **for** $j$ in $1, \ldots, k$ **do**
  4:         Compute $\mathrm{s}(\cdot; \mathcal{G}', f_j, b_j)$ for all nodes using equations 2–3
  5:         Compute $\psi(\cdot; \mathcal{G}', f_j, b_j, h)$ for all nodes using equations 5–6
  6:         Compute $\rho(\cdot; \mathcal{G}', f_j, b_j, h)$ for all nodes using equations 7–8
  7:     **end for**
  8:     $E \leftarrow$ all tip nodes in $\mathcal{G}'$
  9:     Terminate early if $|E| = 0$ OR $\mathrm{s}(m_t; \mathcal{G}', f_j, b_j) = 1 \, \forall j$
10:     $m_e \leftarrow \arg \max_{m \in E} \alpha(m; \mathcal{G}', \xi_f, \xi_b, h)$ (cf. equation 9, breaking ties arbitrarily)
11:     Add all reactions and molecules from $B(m_e)$ to $\mathcal{G}'$
12: **end for**
13: **return** $\mathcal{G}'$
---

successful synthesis plans *in scenarios where all existing synthesis plans currently fail*. Specifically, we propose to choose molecules by maximizing the objective

$$\alpha(m; \mathcal{G}', \xi_f, \xi_b, h) = \mathbb{E}_{f \sim \xi_f, b \sim \xi_b} \left[ \mathbf{1}_{\mathrm{s}(m_t; \mathcal{G}', f, b) = 0} \left[ \rho(m; \mathcal{G}', f, b, h) \right] \right] , \qquad (9)$$

which is proportional to the expected value of $\rho(m)$ conditioned on $\mathrm{s}(m_t) = 0$. We call the resulting algorithm *retro-fallback* (from "retrosynthesis with fallback plans") and state it explicitly in Algorithm 1. The sections are colour-coded for clarity. After initializing $\mathcal{G}'$ to just the target molecule, the algorithm performs $L$ iterations of expansion (although this termination condition could be changed as needed). In each iteration, first the values of s, $\psi$, and $\rho$ are computed for each sample.[5] Next, the algorithm checks whether there are no nodes to expand or whether the root molecule is synthesized for every sample, and if so terminates (both of these conditions mean no further improvement is possible). Finally, a tip node maximizing equation 9 is chosen and used to expand $\mathcal{G}'$.

# 5 Related Work

Mechanistically, retro-fallback most closely resembles retro* (Chen et al., 2020): both perform a bottom-up and top-down update to determine the value of each potential action and select actions greedily. In fact, if costs are defined to be negative log probabilities then the updates for $\psi$ and $\rho$ are essentially equivalent to the "reaction number" and "retro* value" updates from (Chen et al., 2020). The key difference is that retro-fallback performs parallel updates using many samples from $\xi_f$ and $\xi_b$ and combines information from all samples to make a decision, while retro* uses only a single cost. This difference is what allows retro-fallback to directly optimize SSP, while retro* cannot. This ability also distinguishes retro-fallback from other search algorithms such as MCTS (Segler et al., 2018) and proof-number search (Kishimoto et al., 2019).

Works outside of multi-step planning are only tangentially related. Works proposing search heuristics for retrosynthesis search algorithms complement rather than compete with our work: such heuristics could also be applied to retro-fallback. Generative models to produce synthesis plans effectively also function as heuristics. Finally, methods to predict individual chemical reactions are sometimes also referred to as "retrosynthesis models". Retro-fallback solves a different problem: it is a *multi-step* search algorithm which would *use* a reaction prediction model to define the search graph.

---

[5]This order is chosen because s depends only on $f$ & $b$, $\psi$ depends on s, and $\rho$ depends on $\psi$. Because the optimal algorithm to compute s, $\psi$, $\rho$ may depend on $\mathcal{G}'$, we only specify this computation generically.

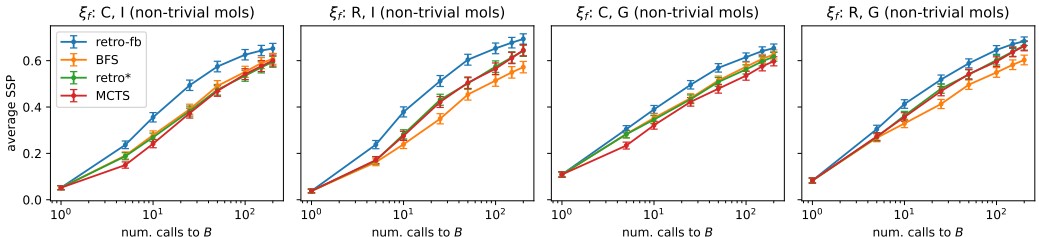

Figure 3: Results with optimistic heuristic on "non-trivial" molecules. C and R refer to constant and rank marginal probabilities, while I and G refer to independent and GP-induced correlations. Solid lines are sample means (averaged across molecules), with error bars representing standard errors.

# 6 Experiments

## 6.1 Experiment Setup

We test retro-fallback on four retrosynthesis tasks using four different feasibility models. We base all of our feasibility models on the pre-trained template classifier from Chen et al. (2020) restricted to the top-50 templates. We vary our feasibility model across two axes: the *marginal* feasibility assigned to each reaction and the *correlation* between feasibility outcomes. Marginally, we consider a constant value (C) and a value which decreases with the rank (R) for marginal feasibility. For correlations we consider all outcomes being independent (I) or determined by a latent GP model (G) which positively correlates similar reactions. Details of these models are given in Appendix C.1.1. Analogous to Chen et al. (2020), we create a buyability model based on eMolecules library, although we use the September 2023 version and exclude molecules with excessively long shipping times (details in Appendix C.1.2).

We compare retro-fallback to breadth-first search (an uninformed search algorithm) and heuristic-guided algorithms retro* (Chen et al., 2020) and MCTS (Segler et al., 2018; Genheden et al., 2020; Coley et al., 2019b). MCTS and retro* were re-implemented and adapted to maximize SSP, which most notably entailed replacing costs or rewards from the backward reaction model $B$ with quantities derived from $\xi_f$ and $\xi_b$ (see Appendix C.1.4 for details) and standardizing their search heuristics. All algorithms are run with a budget of 200 calls to $B$. The presence of heuristics makes comparing algorithms difficult because the choice of heuristic will strongly influence an algorithm's behaviour. We decided to use an *optimistic* heuristic and a heuristic based on the synthetic accessibility (SA) score (Ertl and Schuffenhauer, 2009), which has been shown to be a good heuristic for retrosynthesis in practice despite its simplicity (Skoraczyński et al., 2023).

We test all algorithms on 500 molecules randomly selected from the GuacaMol test set (Brown et al., 2019), which contains drug-like molecules known to be synthesizable, but with a wide range of difficulties (details in Appendix C.1.3). Our primary evaluation metric is the SSP values estimated with $k = 10\,000$ samples, averaged over the 500 molecules.

## 6.2 How effective is retro-fallback?

Since retro-fallback is designed to maximize SSP, the most basic question is whether it does so more effectively than other algorithms. We found that a minority of molecules are "trivial", and all algorithms achieve a SSP of $\approx 1$ within a few iterations. In Figure 3 we plot the average SSP for all "non-trivial" test molecules as a function of number of reaction model calls using an optimistic heuristic for all feasibility models. Retro-fallback clearly outperforms the other algorithms in all scenarios by a wider margin than the error bars. The difference is larger for the feasibility models with independent reactions than with GP-correlated reactions. We suspect this is because there are many synthesis plans with similar reactions: when reaction outcomes are uncorrelated these synthesis plans act as backup plans for each other, but not when they are correlated. The same trends can be seen when using the SA heuristic function (shown in Figure C.1). Overall, this result shows us what

we expect: that retro-fallback maximizes the metric it was specifically designed to maximize more effectively than baseline algorithms.

A natural follow-up question is whether retro-fallback also performs well by metrics other than SSP. In Figures C.2–C.3 we show that for both the optimistic and SA score heuristics retro-fallback is able to find potential solutions for more molecules and produce "best" solutions whose quality closely matches other algorithms. This suggests that it functions as an effective search algorithm even just considering the metrics from past papers.

# 7 Discussion, Limitations, and Future Work

In this paper we presented a novel evaluation metric for retrosynthesis algorithms called "successful synthesis probability" (SSP), proposed a novel algorithm called retro-fallback to greedily maximize SSP, and showed experimentally that retro-fallback is more effective than previously-proposed algorithms. Together, these contributions compensate for the limited ability of existing algorithms to explicitly account for reaction failure and propose backup plans.

One challenge for deploying retro-fallback in practice is the lack of established feasibility and buyability models. To our knowledge, retro-fallback is the first algorithm which can fully utilize uncertainty on the *function* level, so it is not surprising that not much work has been done in this area before. We therefore do not view this as a limitation of our work, but rather as motivation for subsequent research into quantifying the uncertainty of reaction models (especially by domain experts).

Our contributions also have some important conceptual limitations. First, chemists care about the cost and length of synthesis plans in addition to whether they will work, and we do not see a way to incorporate these directly into either our stochastic process formalism or retro-fallback. Second, while our definition of SSP considers an arbitrary number of plans, in practice chemists are unlikely to try more than $\approx 10$ plans before moving on to something else. Nonetheless, all algorithms make assumptions which are untrue, and we are optimistic that the assumptions made by retro-fallback are sufficiently benign that the algorithm can still be useful in practice.

Finally, since retro-fallback uses a search heuristic there is potential to learn this heuristic using the results of past searches ("self-play"). We naturally expect this to improve performance and view this as an exciting direction for future work.

## Acknowledgments and Disclosure of Funding

Thanks to Katie Collins for proofreading the manuscript and providing helpful feedback. Austin Tripp acknowledges funding via a C T Taylor Cambridge International Scholarship and the Canadian Centennial Scholarship Fund. José Miguel Hernández-Lobato acknowledges support from a Turing AI Fellowship under grant EP/V023756/1.

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

# A  Summary of Notation

Although we endeavoured to introduce all notation in the main text of the paper in the section where it is first used, we re-state the notation here for clarity.

## General Math

| | |
|---|---|
| $2^S$ | Power set of set $S$ (set of all subsets of $S$) |
| $\mathbf{1}_{\text{event}}$ | Indicator function: 1 if "event" is True, otherwise 0 |
| $O(N^p)$ | Big-O notation (describing scaling of an algorithm) |
| $\tilde{O}(N^p)$ | Big-O notation, omitting poly-logarithmic factors (e.g. $O(N \log N)$ is equivalent to $\tilde{O}(N)$) |

## Molecules and reactions

| | |
|---|---|
| $m$ | a molecule |
| $r$ | a reaction (assumed to be single-product) |
| $\mathcal{M}$ | space of molecules |
| $\mathcal{R}$ | space of reactions |
| $\mathcal{I}$ | Inventory of buyable molecules |
| $B$ | backward reaction model |

## Search Graphs

| | |
|---|---|
| $\mathcal{G}$ | *implicit* search graph with molecule (OR) nodes in $\mathcal{M}$ and reaction (AND) nodes in $\mathcal{R}$ |
| $\mathcal{G}'$ | *explicit* graph stored and expanded for search. $\mathcal{G}' \subseteq \mathcal{G}$ |
| $Pa_{\mathcal{G}'}(x)$ | The parents of molecule or reaction $x$ in $\mathcal{G}'$. |
| $Ch_{\mathcal{G}'}(x)$ | The children of molecule or reaction $x$ in $\mathcal{G}'$. |
| $T$ | A synthesis plan in $\mathcal{G}$ or $\mathcal{G}'$ |

## Feasibility and Buyability

| | |
|---|---|
| $f$ | Feasible function (assigns whether a reaction is feasible) |
| $b$ | Buyable function (assigns whether a molecule is buyable) |
| $\xi_f$ | feasibility stochastic process (distribution over $f$) |
| $\xi_b$ | buyability stochastic process (distribution over $b$) |
| $\text{s}(m; \mathcal{G}', f, b)$ | Whether a molecule is synthesizable using reactions/starting molecules in $\mathcal{G}'$, with feasible/buyable outcomes given by $f, b$. Takes values in $\{0, 1\}$. Defined in equations 2–3 |
| $\text{s}(m)$ | Shorthand for $\text{s}(m; \mathcal{G}', f, b)$ when $\mathcal{G}', f, b$ are clear from context. |
| $\bar{\text{s}}(m; \mathcal{G}', \xi_f, \xi_b)$ | Expected value of $\text{s}(m; \mathcal{G}', f, b)$ when $f \sim \xi_f, b \sim \xi_b$. Defined in equation 1. |
| $\bar{\text{s}}(m)$ | Shorthand for $\bar{\text{s}}(m; \mathcal{G}', \xi_f, \xi_b)$ when $\mathcal{G}', \xi_f, \xi_b$ are clear from context |
| $\hat{\text{s}}(m; \mathcal{G}', \xi_f, \xi_b)$ | Estimate of $\bar{\text{s}}(m)$ from samples (equation 4) |

## Retro-fallback

| | |
|---|---|
| $m_t$ | the target molecule (at the root of $\mathcal{G}'$) |
| $h$ | Search heuristic function $\mathcal{M} \mapsto [0, 1]$ |
| $\psi(m; \mathcal{G}', f, b, h)$ | Estimate of potential $s(m)$ if one plan under $m$ is expanded. Defined in equations 5–6 |
| $\rho(m; \mathcal{G}', f, b, h)$ | Estimate of potential for $s(m_t)$ if one plan under $m$ is expanded. Defined in equations 7–8 |

We also use the following mathematical conventions throughout the paper:

- $\log 0 = -\infty$
- $\max_{x \in \emptyset} f(x) = -\infty$ (the maximum of an empty set is always $-\infty$)

# B Proofs and Theoretical Results

This appendix contains proofs of theoretical results from the paper.

## B.1 Computing SSP is NP-hard

First, we formally state this result as a theorem:

**Theorem B.1.** *Unless $P = NP$, there does not exist an algorithm to compute $\bar{s}(m_t; \mathcal{G}', \xi_f, \xi_b)$ for arbitrary $\xi_f, \xi_b$ whose time complexity grows polynomially with the number of nodes in $\mathcal{G}'$.*

Theorem B.1 is a corollary of the following theorem, which we prove below.

**Theorem B.2.** *Unless $P = NP$, there does not exist a polynomial time algorithm to determine whether $\bar{s}(m_t; \mathcal{G}', \xi_f, \xi_b) > 0$ whose time complexity grows polynomially with the number of nodes in $\mathcal{G}'$ for arbitrary $\xi_f, \xi_b$.*

Note that Theorem B.2 is distinct from Theorem B.1: the latter is a hardness result about computing SSP, while the former considers only the binary problem of determining whether SSP is zero or non-zero. We now state a proof of Theorem B.2:

*Proof.* We will show a reduction from the Boolean 3-Satisfiability Problem (3-SAT) to the problem of determining whether SSP is non-zero. As 3-SAT is known to be NP-hard (Karp, 1972), this will imply the latter is also NP-hard, completing the proof.

To construct the reduction, assume an instance $I$ of 3-SAT with $n$ variables $x_1, ..., x_n$, and $m$ clauses $c_1, ..., c_m$, each $c_j$ consisting of three literals (where a literal is either a variable or its negation) We will construct an AND-OR graph $\mathcal{G}(I)$ with size $O(n + m)$, alongside with distributions $\xi_f(I)$ and $\xi_b(I)$, such that the SSP in the constructed instance is non-zero if and only if $I$ is satisfiable.

In our construction we first set $\xi_f \equiv 1$, i.e. assume all reactions described below are always feasible.

We then construct a set $P$ of $2n$ potentially buyable molecules, corresponding to variables $x_i$ as well as their negations $\neg x_i$; to simplify notation, we will refer to these molecules as $x_i$ or $\neg x_i$. We then set $\xi_b(I)$ to a uniform distribution over all subsets $S \subseteq P$ such that $\forall_i |S \cap \{x_i, \neg x_i\}| = 1$; in other words, either $x_i$ or $\neg x_i$ can be bought, but never both at the same time. Note that with this construction it is easy to support all necessary operations on $\xi_b$, such as (conditional) sampling or computing marginals.

It remains to translate $I$ to $\mathcal{G}(I)$ in a way that encodes the clauses $c_j$. We start by creating a root OR-node $r$, with a single AND-node child $r'$. Under $r'$ we build $m$ OR-node children, corresponding to clauses $c_j$; again, we refer to these nodes as $c_j$ for simplicity. Finally, for each $c_j$, we attach 3 children, corresponding to the literals in $c_j$. Intuitively these 3 children would map to three molecules from the potentially buyable set $P$, but formally the children of $c_j$ should be AND-nodes (while $P$ contains molecules, i.e. OR-nodes); however, this can be resolved by adding dummy single-reactant reaction nodes.

To see that the reduction is valid, first note that $r$ is synthesizable only if all $c_j$ are, which reflects the fact that $I$ is a binary AND of clauses $c_j$. Moreover, each $c_j$ is synthesizable if at least one of its 3 children is, which translates to at least one of the literals being satisfied. Our construction of $\xi_b$ allows any setting of variables $x_i$ as long as it's consistent with negations $\neg x_i$. Taken together, this means the SSP for $\mathcal{G}(I)$ is non-zero if and only if there exists an assignment of variables $x_i$ that satisfies $I$, and thus the reduction is sound. $\qquad\square$

**Corollary B.3.** *If a polynomial time algorithm did exist to compute the exact value of $\bar{s}(m_t; \mathcal{G}', \xi_f, \xi_b)$, this algorithm would clearly also determine whether $\bar{s}(m_t; \mathcal{G}', \xi_f, \xi_b) > 0$ in polynomial time, violating Theorem B.2. This proves Theorem B.1.*

## B.2 Computing "success" quantities in polynomial time

Retro-fallback, and more generally the calculation of SSP rely on solving a system of recursively-defined equations: equations 2–3 for s, equations 5–6 for $\psi$, and equations 7–8 for $\rho$. Exactly how these equations are solved is detached from the actual algorithms: all that matters is that they are

solved. Depending on the structure of $\mathcal{G}'$, different algorithms with different scaling properties may be applicable. Currently we are uncertain about what the overall "best" algorithm is, and therefore do not advocate for a particular method in this paper. In this section we merely aim to prove a minimal result: that these quantities can always be computed in polynomial time (with respect to the size of the graphs).

First, we state a general theorem applicable to all graphs.

**Theorem B.4.** *Let $|\mathcal{G}'| = N$ (i.e. $\mathcal{G}'$ has N nodes) and that the number of outgoing edges from any node is at most $K < N$. There exists an algorithm with $\tilde{O}(N^2)$ time complexity to compute* s, $\psi$, *and* $\rho$.

*Proof.* Our proof builds on a result from Chakrabarti (1994) which gives an algorithm to compute minimum costs in an algorithm called AO*, which performs minimum-cost search on AND/OR graphs. Let $c_t(n)$ denote the *terminal* cost of a node (analogous to its purchase cost). This will generally be $\infty$ for nodes which are non-terminal (e.g. non-purchasable molecules), and a non-negative real number otherwise. Let $c^*$ denote the optimal cost of a node, and $c_e$ denote the edge cost between two nodes. AO* defines the following cost function:

$$c^*(n) = \min \left[ c_t(n), \min_{n' \in Ch_{\mathcal{G}'}(n)} [c^*(n') + c_e(n, n')] \right] \quad \text{(OR node)} \tag{10}$$

$$c^*(n) = \sum_{n' \in Ch_{\mathcal{G}'}(n)} [c^*(n') + c_e(n, n')] \quad \text{(AND node)} \tag{11}$$

Chakrabarti (1994) presents an algorithm called `Iterative_revise` whose worst case time complexity is $\tilde{O}(N^2)$. Critically, unlike previous algorithms for AO*, the algorithm from Chakrabarti (1994) does *not* assume a tree or acyclic graph, making it very general. Our proof strategy is to transform the equations for s, $\psi$, and $\rho$ to resemble equations 10–11, making `Iterative_revise` applicable and proving our result.

First, define $s'(\cdot; \mathcal{G}', f, b) = -\log s(\cdot; \mathcal{G}', f, b)$. The resulting recursive equations are:

$$s'(m; \mathcal{G}', f, b) = \min \left[ -\log b(m), \min_{r \in Ch_{\mathcal{G}'}(m)} s'(r; \mathcal{G}', f, b) \right]$$

$$s'(r; \mathcal{G}', f, b) = -\log f(r) + \sum_{m \in Ch_{\mathcal{G}'}(r)} s'(m; \mathcal{G}', f, b)$$

Setting $c_t(m) = -\log b(m)$, $c_e(m, r) = 0$, and $c_e(r, m) = -\log f(r)$ these equations clearly match equations 10–11, and the convention that $\log 0 = -\infty$ the costs are guaranteed to be non-negative, making the result applicable. The same transformation and correspondence can be achieved for $\psi$ by defining $\psi'(\cdot; \mathcal{G}', f, b, h) = -\log \psi(\cdot; \mathcal{G}', f, b, h)$ and defining $c_t(m) = -\log \max [b(m), h(m)]$.

Second, define $\rho'(\cdot; \mathcal{G}', f, b, h) = \log \frac{\psi(m_t; \mathcal{G}', f, b, h)}{\rho(\cdot; \mathcal{G}', f, b, h)}$. This results in the recursive equations:

$$\rho'(m; \mathcal{G}', f, b, h) = \begin{cases} 0 & m = m_t \\ \min_{r \in Pa_{\mathcal{G}'}(m)} \rho'(r; \mathcal{G}', f, b, h) & \text{all other } m \end{cases}$$

$$\rho(r; \mathcal{G}', f, b, h) = \begin{cases} \infty & \psi(r; \mathcal{G}', f, b, h) = 0 \\ \rho'(Pa_{\mathcal{G}'}(r)) + \log \frac{\psi(Pa_{\mathcal{G}'}(r))}{\psi(r)} & \psi(r; \mathcal{G}', f, b, h) > 0 \end{cases}$$

If the directions of all edges are flipped (so parents become children and children become parents), then these equations correspond to equations 10–11 with $c_t(n) = 0$ for $m_t$ only, $c_e(m, r) = 0$, and $c_e(r, m) = \log \frac{\psi(m)}{\psi(r)}$ (which is non-negative because of the $\max$ in equation 5).

This completes the proof for all three quantities. $\qquad \square$

Theorem B.4 assumes that the number of outgoing edges in each node is bounded. This is a realistic assumption in retrosynthesis: most reactions involve 1–2 reactants. Reactions with 3 or more reactants are less common, and more than $\approx 10$ is essentially unheard of. Although there may be a large number of possible reactions that can be done on a given molecule, a backward reaction model $B$

usually limits the number of reactions which are added to the graph. Many previous works have used a limit of 50 (Segler et al., 2018; Chen et al., 2020). Therefore we think this assumption is realistic in practice.

One implication of Theorem B.4 is that `Iterative_revise` could be used to directly compute $s, \psi,$ and $\rho$. However, in some cases this is likely sub-optimal: for example, if $\mathcal{G}'$ is acyclic then these quantities can be computed in linear time using a single pass over all nodes. Although the presence of reversible reaction (e.g. $A \rightarrow B$ and $B \rightarrow A$) make it unlikely that strictly acyclic graphs will be encountered in practice during retrosynthesis, cyclic plans will *not* yield optimal plans we expect very few cycles to be explored in $\mathcal{G}'$. Therefore we propose in practice to initialize $s, \psi, \rho$ to 0 and then iterate their recursive relations until convergence. At this time we do not have any proofs for the time complexity of this procedure, but in practice it appears to be sub-linear. However, in Collorary B.5 we show that computing $\rho$, which is the last phase of the algorithm, can indeed be done in linearithmic time. As this optimization is not applicable to computing $\psi$, it does not improve the overall complexity of the algorithm.

**Corollary B.5.** *Let $|\mathcal{G}'| = N$ (i.e. $\mathcal{G}'$ has $N$ nodes) and that the number of outgoing edges from any node is at most $K < N$. There exists an algorithm with $O(N \log N)$ time complexity to compute $\rho$ from $\psi$.*

*Proof.* Recall the reduction of computing $\rho$ to minimum-cost search in an AND/OR graph from the proof of Theorem B.4. Note that the AND nodes in the resulting graph always have *at most one child* (corresponding to the node parent in the original tree), thus the sum-over-children component seen for AND nodes in general AND/OR graph search does not appear. Consequently, it is easy to see this particular search problem is equivalent to finding a shortest path from $m_t$ to every other node, which can be done using Dijkstra's algorithm in $O(N \log N)$ time. □

## B.3 Errors for estimating Bernoulli random variables

Errors of i.i.d random variables are well-studied. The rate at which the sample mean "concentrates" around its expected value can be bounded using any number of concentration bounds. For example, applying the Chernoff bound using $k = 10\,000$ yields:

$$P\left(|\hat{s}(m; \mathcal{G}', \xi_f, \xi_b, k) - \bar{s}(m; \mathcal{G}', \xi_f, \xi_b)| > 0.025\right) < 10^{-5} \quad \forall \mathcal{G}', \xi_f, \xi_b$$

This means that with $10\,000$ samples, the SSP can be placed within a 5% interval with near-certainty in all settings. We believe that a higher level of accuracy is not likely to be useful for chemists: if $\bar{s}(m)$ is reasonably large than a 5% error is relatively small, while if $\bar{s}(m)$ is near zero then a chemist will probably just choose not try to make the molecule (and therefore the distinction between 0.1% and 5% is not actually that important).

## C  Extended experiment section

### C.1  Details of experimental setup

#### C.1.1  Feasibility models

As stated in section 6, we examined four feasibility models for this work, which assign different marginal feasibility values and different correlations between feasibility outcomes. The starting point for our feasibility models was the opinion of a trained organic chemist that around 25% of the reactions outputted by the pre-trained template classification model from Chen et al. (2020) were "obviously wrong". From this, we proposed the following two marginal values for feasibility:

1. (C) A constant value of $1/2$ for all reactions. This is an attempt to account for the 25% of reactions which were "obviously wrong", plus an additional unknown fraction of reactions which seemed plausible but may not work in practice. Ultimately anything in the interval $[0.2, 0.6]$ seemed sensible to use, and we chose $1/2$ as a nice number.

2. (R) Based on previous work with template classifiers suggesting that the quality of the proposed reaction decreases with the softmax value (Segler and Waller, 2017; Segler et al., 2018), we decided to assign higher feasibility values to reactions with high softmax values. To avoid overly high or low feasibility values, we decided to values based on the *rank* of the outputted reaction, designed the following function which outputs a high feasibility ($\approx$75%) for the top reaction and decreases to ($\approx$10%) for lower-ranked reactions:

$$p(\text{rank}) = \frac{0.75}{1 + \text{rank}/10} \ . \tag{12}$$

Note that "rank" in the above equation starts from 0.

We then added correlations on top of these marginal feasibility values. The independent model (I) is simple: reaction outcomes are sampled independently using the marginal feasibility values described above. To introduce some correlations without changing the marginal probabilities, we created the following probabilistic model which assigns feasibility outcomes by applying a threshold to the value of a latent Gaussian process (Williams and Rasmussen, 2006):

$$\text{outcome}(z) = \mathbf{1}_{z>0} \tag{13}$$
$$z(r) \sim \mathcal{GP}\left(\mu(\cdot), K(\cdot, \cdot)\right) \tag{14}$$
$$\mu(r) = \Phi^{-1}\left(p(r)\right) \tag{15}$$
$$K(r, r) = 1 \quad \forall r \tag{16}$$

Here, $\Phi$ represents the CDF of the standard normal distribution and $p(r)$ represents the desired marginal probability function. Because of equation 16, the marginal distribution of each reaction's $z$ value is $\mathcal{N}(\Phi^{-1}(p(r)), 1)$ which will be positive with probability $p(r)$. This ensures consistency with any desired marginal distribution for any kernel $K$ with diagonal values of 1. If $K$ is the identity kernel (i.e. $K(r_1, r_2) = \mathbf{1}_{r_1 = r_2}$) then this model implies all outcomes are independent. However, non-zero off-diagonal values of $K$ will induce correlations (positive or negative).

We aimed to design a model which assigns correlations very conservatively: only reactions involving similar molecules *and* which induce similar changes in the reactant molecules will be given a high positive correlation; all other correlations will be near zero. We therefore chose a kernel as a product of two simpler kernels:

$$K_{\text{total}}(r_1, r_2) = K_{\text{mol}}(r_1, r_2) K_{\text{mech}}(r_1, r_2) \ .$$

We chose $K_{\text{mol}}(r_1, r_2)$ to be the Jaccard kernel $k(x, x') = \frac{\sum_i \min(x_i, x_i')}{\sum_i \min(x_i, x_i')}$ between the Morgan fingerprints (Rogers and Hahn, 2010) with radius 1 of the entire set of product and reactant molecules.[6] We chose $K_{\text{mech}}(r_1, r_2)$ to be the Jaccard kernel of the *difference* between the product and reactant fingerprints individually. This is sensible because the difference between fingerprint vectors corresponds to a set of subgraphs which are added/removed as part of the reaction. Reactions which perform the same kinds of transformation will induce the same kinds of difference vectors.

---

[6]This is the same as adding the fingerprint vectors for all component molecules.

### C.1.2 Buyability Models

Following Chen et al. (2020) we based our buyability models on the inventory of eMolecules: a chemical supplier which acts as a middleman between more specialized suppliers and consumers. According to eMolecule's promotional material they offer 6 "tiers" of molecules:

0. (*Accelerated Tier*). "Delivered in 2 days or less, guaranteed. Most reliable delivery service. Compound price is inclusive of a small service fee, credited back if not delivered on time. Available in the US only."

1. "Shipped within 1- 5 business days. Compounds from suppliers proven to ship from their location in $< 5$ days."

2. "Shipped within 10 business days. Compounds from suppliers proven to ship from across the globe in $< 10$ days"

3. "Shipped within 4 weeks. Shipped from suppliers further from your site and often with more complex logistics. Synthesis may be required using proven reactions."

4. "Shipped within 12 weeks. Usually requires custom synthesis on demand."

5. "Varied ship times. Requires custom synthesis for which a quote can be provided on request."

Much like machine learning researchers, chemists usually want to complete experiments as quickly as possible and probably would prefer not to wait 12 weeks for a rare molecule to be shipped to them. Such molecules could arguably be considered less "buyable" on this subjective basis alone, so we decided to create buyability models based on the tier of molecule. Unfortunately, the public repository for retro* does not contain any information on the tier of each molecule, and because their inventory was downloaded in 2019 this information is no longer available on eMolecules' website. Therefore we decided to re-make the inventory using the latest data.

We downloaded data from eMolecules downloads page[7], specifically their "orderable" molecules and "building blocks" with quotes. After filtering out a small number of molecules (31407) whose SMILES were not correctly parsed by rdkit we were left with 14903392 molecules with their associated purchase tiers. Based on this we created 2 buyability models:

- **Basic:** all molecules in tiers 0-2 are purchasable with 100% probability. Corresponds to realistic scenario where chemists want to do a synthesis and promptly.

- **Complex:** molecules are independently purchasable with probability that depends on the tier (100% for tiers 0-2, 50% for tier 3, 20% for tier 4, 5% for tier 5). These numbers were chosen as subjective probabilities that the compounds would be delivered within just 2 weeks (shorter than the longer times advertised). This still corresponds to a chemist wanting to do the synthesis within 2 weeks, but being willing to risk ordering a molecule whose stated delivery time is longer.

*The experiments in section 6 use only the **basic** buyability model.* We performed some preliminary experiments with the **complex** buyability model but found that in most cases there was no difference. This makes sense: eMolecules is a real, profit-driven company and there is a clear financial incentive to quickly ship molecules which are useful for a wide range of syntheses. Molecules with longer shipping times are used more rarely, so one would expect them to only be useful in a smaller number of cases. Because of the page limit for conference papers, we decided to prioritize other experiments for this manuscript, and therefore do not show any results for this buyability model.

In the future, we believe that better buyability models could be formed by introducing correlations between molecules coming from the same supplier, but we do not investigate that here (chiefly because the eMolecules data we downloaded does not contain information about suppliers).

### C.1.3 Test molecules

The test molecules were generated with the following procedure:

1. Download the publicly available test set from Brown et al. (2019)

---

[7]Downloaded 2023-09-08.

2. Filter our all molecules available in the eMolecules inventory (C.1.2)

3. Shuffle all molecules and take the first 500

Code to reproduce this process, and the entire test set in shuffled order is included in our supplementary material.

We note that although many previous works have evaluated their methods on the 190 molecule test set from Chen et al. (2020), this test set is small and contains only molecules where finding any single synthesis plan is difficult, which only occurs for a small minority of molecules. It was unclear to us whether this would be a good test set: in particular, it is likely that the set of molecules where finding one synthesis plan is hard does not completely overlap with the set of molecules where finding *multiple* synthesis plans is hard. By using a more "typical" set of molecules we avoid this bias.

### C.1.4   Algorithm configuration

Retro-fallback was run with $k = 1000$ samples from $\xi_f, \xi_b$. We configured other algorithms to try to maximize the closest proxy to SSP: In particular, this means:

- Breadth-first search was run with no modifications.

- retro* was run using $-\log \mathbb{E}_f[f(r)]$ as the reaction cost and $-\log \mathbb{E}_b[b(m)]$

- MCTS was run using $\hat{s}(m; T, \xi_f, \xi_b)$ as the reward for finding synthesis plan $T$ (i.e. the empirical SSP for individual synthesis plans). To allow the algorithm to best make use of its budget of reaction model calls, we only expanded nodes after they were visited 10 times. The marginal feasibility value of reach reaction was used as the policy in the upper-confidence bound. We used an exploration constant of $c = 0.01$ to avoid "wasting" reaction model calls on exploration, and only gave non-zero rewards for up to 100 visits to the same synthesis plan to avoid endlessly re-visiting the same solutions.

We chose *not* to compare with proof-number search (Kishimoto et al., 2019) because we did not see a way to configure it to optimize SSP. We chose not to compare with algorithms requiring some degree of learning from self-play due to computational constraints, and because it seemed inappropriate to compare with self-play methods without also learning a heuristic for retro-fallback with self-play.

Because retro-fallback runs on a minimal AND/OR graph, we used a modified version of retro* which also operates on an AND/OR graph. This modified version is not our original creation (it is explained in section 3.5 of Chen et al. (2020)) and is fully consistent with the original tree-based version in that it estimates the same costs and expands the same nodes, it just does not store large duplicate subtrees and uses an alternative shortest-path algorithm to perform updating. We also run breadth-first search on the minimal AND/OR graph (although this requires no special modifications).

### C.1.5   Heuristic functions

The heuristic obviously plays a critical role in heuristic-guided search algorithms! Ideally one would control for the effect of the heuristic by using the same heuristic for different methods. However, this is not possible when comparing algorithms from different families because the heuristics are interpreted differently! For example, in retro-fallback the heuristic is interpreted as a SSP in $[0, 1]$ (higher is better), while in retro* it is interpreted as a cost between $[0, \infty)$ (lower is better). If we used literally the same heuristic it would give opposite signals to both of these algorithms, which is clearly not desirable or meaningful. Therefore, we tried our best to design heuristics which were "as similar as possible."

**Optimistic heuristic**   Heuristics which predict the best possible value are a common choice of naive heuristic. Besides being an important baseline, optimistic heuristics are always *admissible* (i.e. they never overestimate search difficulty), which is a requirement for some algorithms like A* to converge to the optimal solution. For retro-fallback, the most optimistic heuristic is $h_{\mathrm{rfb}}(m) = 1$, while for retro* it is $h_{\mathrm{r*}}(m) = 0$, as these represent the best possible values for SSP and cost respectively. For MCTS, the heuristics is a function of a *partial plan* $T'$ rather than a single molecule. We choose the heuristic to be $\mathbb{E}_{f \sim \xi_f}[\min_{r \in T'} f(r)]$, which is the expected SSP of the plan $T'$ if it were completed

by making every tip molecule buyable.[8] In practice this quantity was estimated from $k$ samples (same as retro-fallback).

**SA score heuristic** SA score gives a molecule a score between 1 and 10 based on a dictionary assigning synthetic difficulties to different subgraphs of a molecule (Ertl and Schuffenhauer, 2009). A score of 1 means easy to synthesize, while a score of 10 means difficult to synthesize. For retro-fallback, we let the estimated SSP decrease linearly with the SA score:

$$h_{\text{rfb}}(m) = 1 - \frac{\text{SA}(m) - 1}{10} \ .$$

Because the reaction costs in retro* were set to negative log feasibility values, we thought a natural extension to retro* would be to use $h_{\text{r*}}(m) = -\log h_{\text{rfb}}(m)$. This choice has the advantage of preserving the interpretation of total cost as the negative log joint probability, which also perfectly matches retro-fallback's interpretation of the heuristic (recall that in section 4.2 the heuristic values were assumed to be independent). We designed MCTS's heuristic to also match the interpretation of "joint probability":

$$h_{\text{MCTS}}(T') = \mathbb{E}_{f \sim \xi_f} \left[ \underbrace{\left( \min_{r \in T'} f(r) \right)}_{\text{reactions feasible}} \prod_{m \in \text{tip}(T'), b(m)=0} h_{\text{rfb}}(m) \right]$$

which is the expected SSP of the plan if all non-purchasable molecules are made purchasable independently with probability $h_{\text{rfb}}(m)$.

### C.1.6 Analysis

Our primary analysis metric was the SSP. For algorithms that use AND/OR graphs (e.g. retro-fallback, retro*), we computed the SSP using equations 2–3 with $k = 10\,000$ samples from $\xi_f, \xi_b$.

For algorithms which use OR trees the best method for analysis is somewhat ambiguous. One option is to extract all plans $T \subseteq \mathcal{G}'$ and calculate whether each plan succeeds on a series of samples $f_i, b_i$. A second option is to convert $\mathcal{G}'$ into an AND/OR graph and analyze it like other AND/OR graphs. Although they seem similar, these options are subtly different: an OR graph may contain reactions in different locations which are not connected to form a synthesis plan, but *could* form a synthesis plan if connected. The process of converting into an AND/OR graph would effectively form all possible synthesis plans which could be made using reactions in the original graph, even if they are not actually present in the original graph. We did implement both methods and found that converting to an AND/OR graph tends to increase performance, so this choice does make a meaningful difference. We think the most "realistic" option is unclear, so for consistency with other algorithms we chose to just convert to an AND/OR graph.

### C.1.7 Software Implementation

Our code is included in the supplementary material of this paper. We built our code around the open-source library SYNTHESEUS[9] and used its implementations of retro* and MCTS in our experiments. The exact template classifier from Chen et al. (2020) was used by copying their code and using their model weights. Our code benefitted from the following libraries:

- `pytorch` (Paszke et al., 2019), `rdkit` and `rdchiral` (Coley et al., 2019a). Used in the template classifier.
- `networkx` (Hagberg et al., 2008). Used to store search graphs and for analysis.
- `numpy` (Harris et al., 2020), `scipy` (Virtanen et al., 2020), and `scikit-learn` (Pedregosa et al., 2011). Used for array programming and linear algebra (e.g. in the feasibility models).

### C.2 Additional plots for section 6.2

See Figures C.1, C.2, C.3. These figures are discussed in section 6.2.

---

[8]Note that the $\min$ function will be 1 if *all* reactions are feasible, otherwise 0. Using $\prod_r$ instead of $\min_r$ would yield the same output.

[9]https://github.com/microsoft/syntheseus/

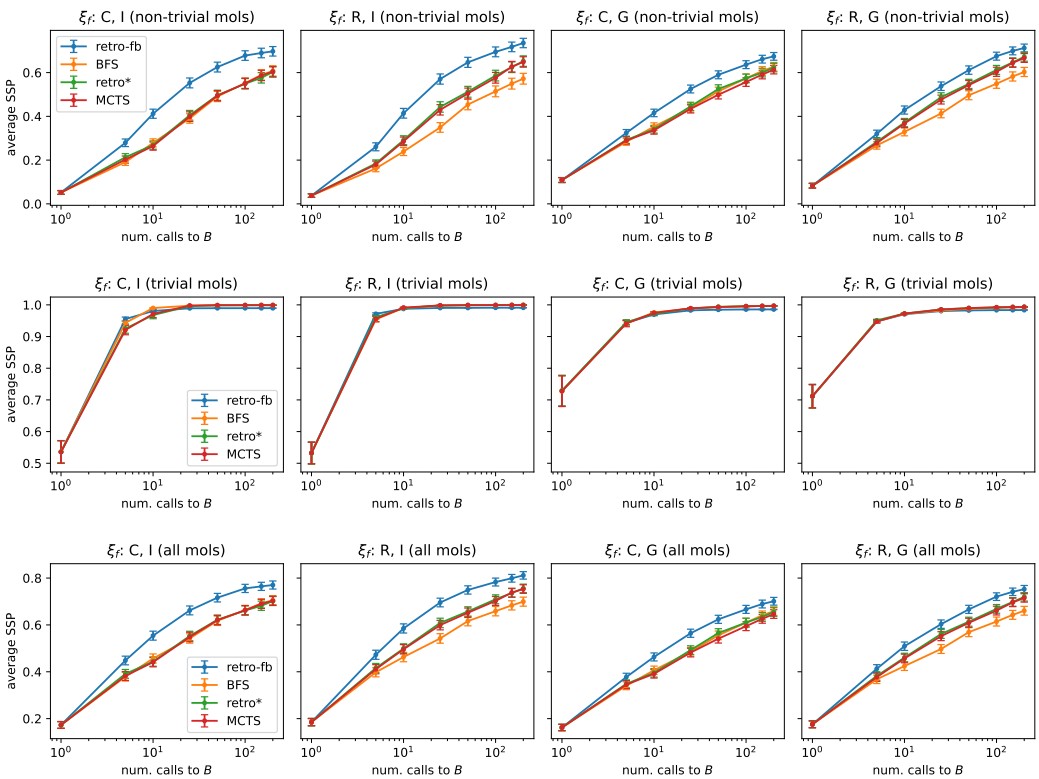

Figure C.1: Average SSP for algorithms using SAscore heuristic (interpretation is the same as Figure 3).

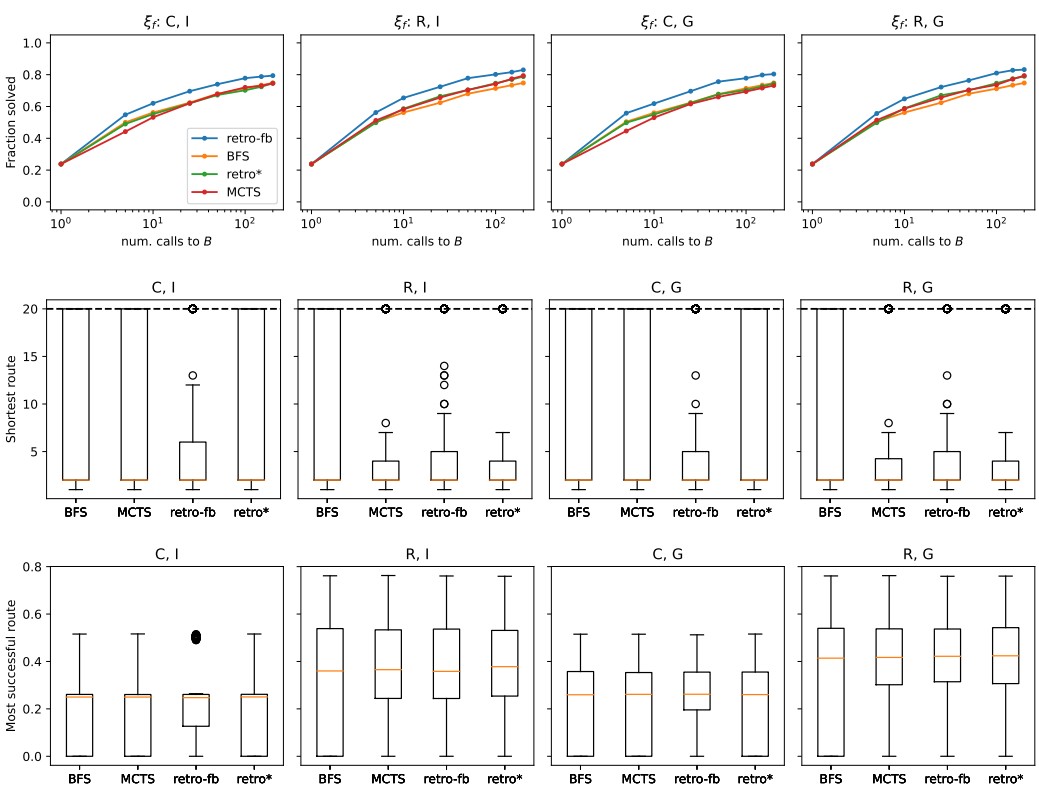

Figure C.2: Alternative success metrics for algorithms with optimistic heuristic.

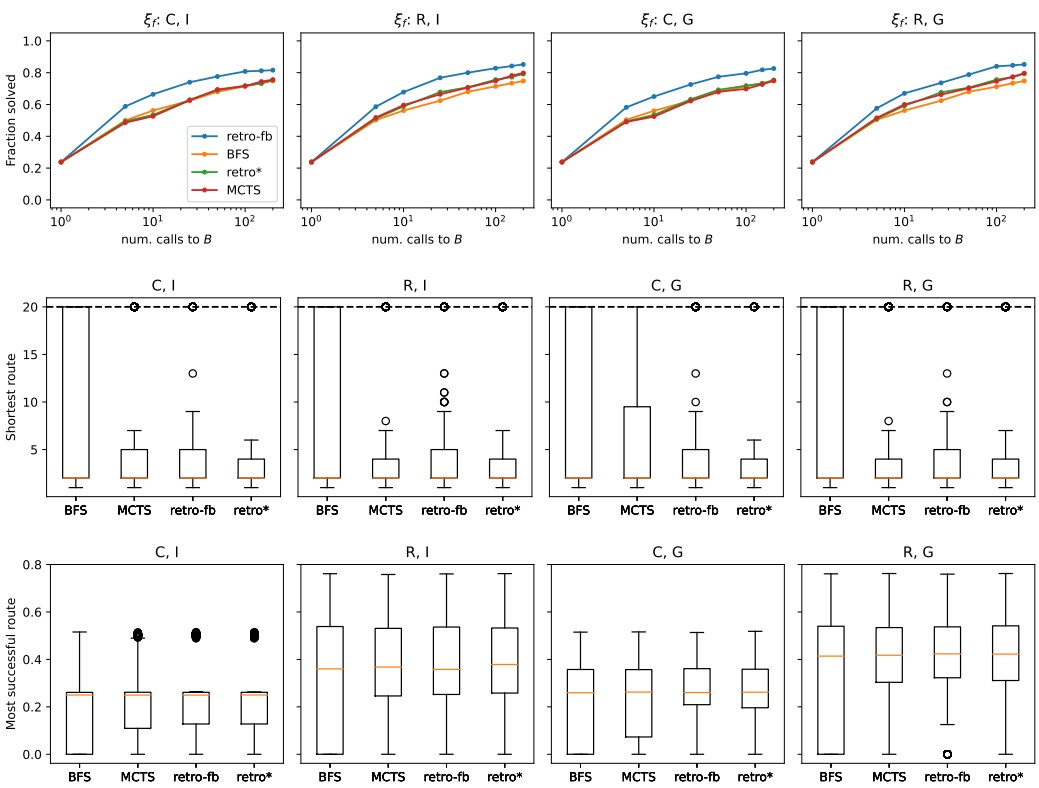

Figure C.3: Alternative success metrics for algorithms with SAscore heuristic.

