# OpenReview forum: "Retro-fallback: retrosynthetic planning in an uncertain world"
_NeurIPS.cc/2023/Workshop/AI4Science — NeurIPS2023-AI4Science Poster_

### Official Review · Reviewer_3jMn · 2023-10-23
**Review of Retro-fallback: retrosynthetic planning in an uncertain world**

**Rating:** 8
**Confidence:** 4

**Review:**

This paper introduces Retro-fallback, an interesting framework that uses stochastic processes to approach the retrosynthesis problem. As far as I know, this is the first work that takes the uncertainty of the reaction allowability in retrosynthesis into account, which is a critical issue in retrosynthesis. Though I think the evaluation is rather simple, a comprehensive evaluation may involve experiments and is beyond the scope of one workshop paper.

---

### Official Review · Reviewer_RKc8 · 2023-10-25
**The paper introduces "retro-fallback," a novel algorithm based on a proposed metric, successful synthesis probability, to enhance the efficiency of searching chemical synthesis pathways and maximize feasibility in the lab.**

**Rating:** 9
**Confidence:** 4

**Review:**

## Pros

* Clear description.
* A very solid piece of paper with a sufficient amount of experiments and discussions.
* The evaluation metric, successful synthesis probability (SSP), is novel and practical. Additionally, the clever use of an and/or tree structure for recursively estimating this value is noteworthy.
* The model's performance significantly surpasses Retro* and BFS on the SSP metric, while also demonstrating its superiority over other models on conventional metrics such as fraction solved, shortest route, and most successful route.

## Cons

* The authors could enhance their discussion of related work and provide a more detailed description of the second part regarding retrosynthesis.

* There may be a distinction between `buyability` and `feasibility`, as in a real-world situation, `buyability` can be determined directly by checking if a molecule is a building block, while `feasibility` might require considering real-world factors. The authors may want to discuss more about the definitions and how to calculate them.

* In Figure F.9, the authors could consider including the `fraction solved`, as it is an important comparative metric.

## Quality

This is a sufficiently good piece of paper, and I learned a lot from it.

## Clarity

Good

## Originality

No issues

---

### Meta-Review · Area_Chair_ZUs6 · 2023-10-26

**Recommendation:** Accept (Poster)
**Confidence:** 5

**Metareview:**

Good paper. Accept.

I suggest the authors consider more metrics such as matching accuracy for retrosynthetic planning to improve this paper.